# A Gaussian Process-Bayesian Bernoulli Mixture Model for Multi-Label Active Learning

**Weishi Shi**[*]    **Dayou Yu** [*]    **Qi Yu**
Golisano College of Computing and Information Sciences
Rochester Institute of Technology
{ws7586,dy2507,qi.yu}@rit.edu

## Abstract

Multi-label classification (MLC) allows complex dependencies among labels, making it more suitable to model many real-world problems. However, data annotation for training MLC models becomes much more labor-intensive due to the correlated (hence non-exclusive) labels and a potentially large and sparse label space. We propose to conduct multi-label active learning (ML-AL) through a novel integrated Gaussian Process-Bayesian Bernoulli Mixture model (GP-B$^2$M) to accurately quantify a data sample's overall contribution to a correlated label space and choose the most informative samples for cost-effective annotation. In particular, the B$^2$M encodes label correlations using a Bayesian Bernoulli mixture of label clusters, where each mixture component corresponds to a global pattern of label correlations. To tackle highly sparse labels under AL, the B$^2$M is further integrated with a predictive GP to connect data features as an effective inductive bias and achieve a feature-component-label mapping. The GP predicts coefficients of mixture components that help to recover the final set of labels of a data sample. A novel auxiliary variable based variational inference algorithm is developed to tackle the non-conjugacy introduced along with the mapping process for efficient end-to-end posterior inference. The model also outputs a predictive distribution that provides both the label prediction and their correlations in the form of a label covariance matrix. A principled sampling function is designed accordingly to naturally capture both the feature uncertainty (through GP) and label covariance (through B$^2$M) for effective data sampling. Experiments on real-world multi-label datasets demonstrate the state-of-the-art AL performance of the proposed model.

## 1  Introduction

In multi-label classification (MLC), each data instance may be associated with more than one label. Such a rich representation of labels can encode more complex data-label distributions that arise in many real-world problems [1, 2, 3, 4]. As a simple yet powerful tool, binary relevance machines (BRMs) transform an MLC problem into multiple binary problems and train independent binary classifiers for each label [5]. Such a transformation gives BRMs the flexibility to leverage state-of-art binary classifiers (e.g., deep neural networks and SVMs). However, applying BRMs to a correlated and potentially large label space poses key challenges. First, many real-world multi-label datasets contain a large number of labels. Training one predictor per label incurs a prohibitive cost. Second, despite an overall large label space, each data instance is usually assigned limited labels. Many labels are relatively rare and their appearances depend not only on the features but also the occurrence of other labels. Predicting these "complex" labels directly using independent binary classifiers is fundamentally difficult due to the limited positive data instances and weaker direct dependency on the features. Correlations among labels provide important auxiliary information to enhance multi-label

---

[*]Equal contribution

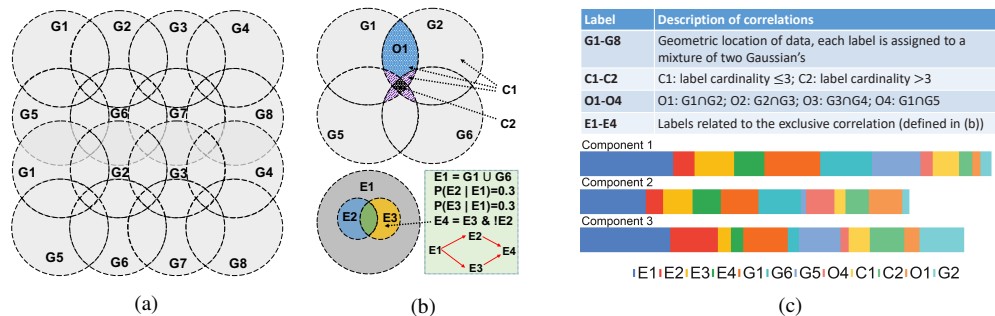

Figure 1: (a) Labels with geometric correlation (G1-G8); (b) Labels with cardinality (C1,C2), overlapping (O1-O4), and exclusive (and hierarchical) dependencies (E1-E4); (c) Definition of label correlation and learned mixture components.

prediction [6, 7, 8]. However, these models heavily rely on the training data that exhibit these important label correlations. Considering the high cost in annotating a multi-label dataset, it is critical to choose the most informative data samples for cost-effective data annotation.

In this paper, we propose a novel **Gaussian Process-Bayesian Bernoulli Mixture (GP-B$^2$M)** model to achieve cost-effective sampling for multi-label active learning (ML-AL). In ML-AL, since labels are not mutually exclusive as in the single label setting, all the labels should be considered collectively when designing an active sampling function so that a data sample's overall contribution to the entire label space can be accurately measured. However, since only limited training data instances are available for an ML-AL model, how to accurately model label correlations and hence *quantify a data sample's overall informativeness using very sparse labels under AL poses a grand challenge*. Existing efforts that explicitly model label correlations usually focus on limited types of correlations such as pairwise [3, 9], conditional [10, 11], or full correlation in a subset of labels [12, 13]. Consequently, those methods may miss some important label correlations. Label correlations can also be captured through a latent embedding [7, 8]. While these methods can scale to a large label space, they usually require a decent number of training labels to compute an accurate embedding, making them less suitable for ML-AL. Furthermore, the learned embedding has no semantic meanings, which cannot be used to interpret the discovered label dependencies [2].

The proposed GP-B$^2$M model addresses the limitations of existing methods to fundamentally advance ML-AL. In particular, the B$^2$M encodes label correlations using a Bayesian Bernoulli mixture of label clusters. Since labels are highly sparse in ML-AL, a predictive GP is further integrated to learn a distribution of mixture coefficients that connect data features with the label clusters. Thus, the label clusters can be regarded as a global pattern of label co-occurrences discovered from both the training labels and data features to address label sparsity. In this novel feature-component-label mapping, data features serve as an *inductive bias* to learn accurate mixture components of labels, where data samples with similar features should be mapped to similar mixture components, which in turn lead to a similar set of labels. Such an inductive bias allows the discovery of label relationships from limited labels with the support of feature relationships, which is essential for a sparse label space in ML-AL.

Figure 1 shows three mixture components learned from synthetic data designed with complex label correlations, including geometric, cardinality, overlapping, and hierarchical (see Figure 1 (c) for definitions). For example, the hierarchical labels ($E1 - E4$) show some interesting but quite complicated correlations that may exist in many real-world data. If $E1$ represents a common disease and $E2, E3$ represent some less common ones that may co-occur with $E1$ (30% of the time), then $E4$ corresponds to a rather rare disease that only co-occurs with $E3$ but not $E2$. In Figure 1 (c), both components 1 and 2 show a high chance of $E4$. It is also clear that while both $E1$ and $E3$ are very likely to appear in these components, the chance of seeing $E2$ is much lower, reflecting its exclusive relationship with $E3$. Interestingly, while both components 1 and 2 cover $E4$, they also show complementary information. In component 1, $E4$ mostly appears in the non-overlapping regions, which is indicated by smaller $O1$ and $O4$; whereas in component 2, it's more likely to

---

[2]Correlation sometimes refers to the degree to which a pair of random variables are linearly related. However, in the broadest sense, both correlation and dependence refer to any statistical relationship between two random variables, so they are used exchangeably in the rest of the paper.

occur in overlapping regions $O1$ and $O4$. Finally, in component 3, it is less likely to observe $E4$. Consequently, the chance to see $E2$ is significantly higher.

The above example demonstrates that the learned mixture components accurately capture complex label correlations that are critical for active data sampling. They are also interpretable, which can help to unveil important relationships among labels. Our **key contribution** is threefold: (i) We propose a novel GP-Bayesian Bernoulli mixture model to discover meaningful label correlations from limited labels by encoding the inductive bias from data features as Bayesian priors to learn from both labels and features while ensuring consistency. (ii) We introduce a set of auxiliary latent variables to achieve a fully conjugate feature-component-label mapping in the Bayesian model to support efficient end-to-end posterior inference. The number of mixture components is also dynamically adjusted during Bayesian inference so that the model complexity is automatically calibrated according to the size of training data, which is critical for AL. (iii) The model outputs the predictive distribution that provides both the label prediction and their correlations in the form of a label covariance matrix. We design a novel active sampling function that integrates both feature uncertainty and label covariance to quantify a data sample's overall contribution to a correlated label space. Extensive experiments on both synthetic and real-world multi-label data and comparison with competitive models demonstrate that the proposed GP-B$^2$M achieves the state-of-the-art active learning performance.

## 2 Related Work

Due to the wide adoption of BRMs for MLC, a number of AL models have been developed based upon BRMs. For example, the estimated reduction of a BRM loss function has been used as an uncertainty criterion for data sampling [14]. Uncertainty from individual SVMs in BRMs has also been integrated to compute a sampling score, where label correlation is used to reduce the complexity of the active query rather than improve active sampling [15]. Label inconsistency provides an alternative way to incorporate label correlation into BRMs [16] for data sampling. This has been further extended through label ranking [17]. As discussed earlier, AL models built upon BRMs do not systematically capture label correlations, which may lead to inaccurate uncertainty measures for ML-AL.

A few existing models capture label correlations explicitly or through a latent embedding to support active multi-label sampling. For example, the approximate entropy of the predicted labels from a Bernoulli Mixtures model (CBM) is used for data sampling [18]. However, the quality of the uncertainty estimation relies on an external multi-class classifier used to predict the component coefficients. Both model selection and parameter tuning for the classifier make it difficult for AL. One fundamental limitation is that CBM was originally designed for MLC (instead of AL) by predicting a distinct set of label clusters for each data sample [6]. Thus, different from the proposed GP-B$^2$M model, no global label clusters are discovered to capture the label correlations, making it unsuitable for multi-label AL. Gao et.al propose a correlation aware active sampling method for transfer learning task [19]. However, the method uses a kernel function to measure the label similarity between two data instances which cannot work well for a large and sparse label space. Compressed sensing (CS) has been employed to learn a latent embedding of the label space to capture potential label correlations, which can then be used to design a sampling function. However, since the latent space is continuous, the labels are further assumed to be drawn from a Gaussian distribution to ensure conjugacy, which violates the binary nature of the labels [20]. Furthermore, CS requires an additional step to recover the predicted label from the latent code, which is less efficient for AL. Active sampling is also sensitive to the recovery process and the recovery quality is usually low at the beginning of AL due to the lack of training data [21].

The proposed GP-B$^2$M model systematically addresses the key limitations of existing methods through a well integrated Bayesian framework that supports a fully conjugate feature-component-label mapping and end-to-end posterior inference for cost-effective ML-AL.

## 3 GP-B$^2$M for Multi-Label Active Learning

In this section, we first describe the proposed GP-B$^2$M model by introducing key latent variables along with their conditional dependencies. We then present a novel posterior inference process by augmenting the original model using a set of auxiliary variables to resolve the non-conjugate prior and likelihood. A principled sampling function is introduced in the end for cost-effective ML-AL.

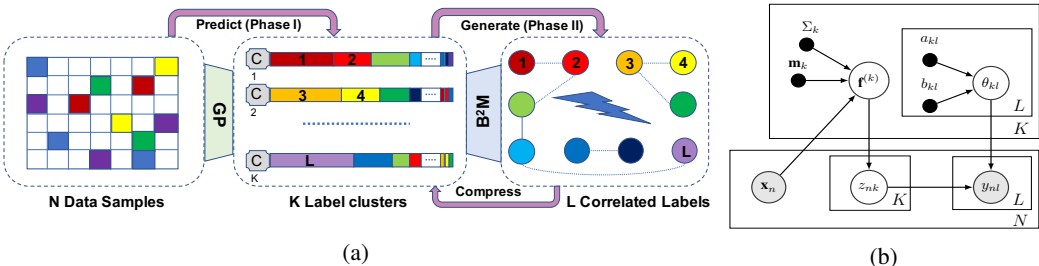

Figure 2: (a) The GP-B$^2$M framework for ML-AL; (b) Graphical model of GP-B$^2$M.

### 3.1 The Bayesian Bernoulli Mixture Model

Let $\mathbf{X} = \{\mathbf{x}_1, ..., \mathbf{x}_N\}$ denote a training set with $N$ data samples and $\mathbf{Y} = \{\mathbf{y}_1, ..., \mathbf{y}_N\}$ denote the labels, where $\mathbf{y}_n \in \{0, 1\}^L$. The proposed GP-B$^2$M model assumes there are $K$ mixture components, $\Theta = \{\boldsymbol{\theta}_k\}_{k=1}^K$, shared by all the data samples and the label vector of each sample is generated from a mixture process. Mixture component $k$ is a result of $L$ Bernoulli experiments governed by two parameters $a_{kl}$ and $b_{kl}$: $\boldsymbol{\theta}_k \sim \prod_{l=1}^L \text{Beta}(a_{kl}, b_{kl})$, where $\theta_{kl}$ denotes the probability of assigning label $l$ to component $k$. The indicator variable $z_{nk}$ denotes whether component $k$ is assigned to sample $\mathbf{x}_n$, where $z_{nk} \sim \text{Cat}(\boldsymbol{\pi}_n)$, $\pi_{nk} = h^{(k)}(\mathbf{f}_n)$, $h^{(k)}$ is a mapping function that outputs the probability of assigning $\mathbf{x}_n$ to mixture component $k$, and $\mathbf{f}_n = (f_n^{(1)}, ..., f_n^{(K)})^T$ are GP latent functions for sample $\mathbf{x}_n$ with $f_n^{(k)} = f^{(k)}(\mathbf{x}_n)$. The final label vector associated with $\mathbf{x}_n$ is drawn from the mixture distribution given by $p(\mathbf{y}_n|\Theta) = \sum_{k=1}^K \pi_{nk}\boldsymbol{\theta}_k$. Figure 2(b) shows the graphical model of this generative process.

The GP-B$^2$M model essentially adopts a **two-phase learning process**, where these two phases are seamlessly integrated (see Figure 2(a)). In phase I, it predicts the probabilistic assignment of the mixture components by learning a distribution of latent functions: $F = \{\mathbf{f}^{(k)}\}_{k=1}^K$, where $\mathbf{f}^{(k)} = (f_1^{(k)}, ..., f_N^{(k)})^T$. In phase II, these predicted mixture assignments are used to refine the parameters of the beta distributions so that updated mixture components can best recover the true label vectors. One key innovation lies in using the latent indicator variables $Z = \{\mathbf{z}_n\}_{n=1}^N$ to link the feature space with the label mixture components as a way to encode the feature related inductive bias. This is achieved through a mapping function $\mathbf{h}_n = (h_n^{(1)}, ..., h_n^{(K)})^T$, with $h_n^{(k)} = h^{(k)}(\mathbf{f}_n)$:

$$p(z_{nk} = 1|\mathbf{f}_n) = \pi_{nk} = h_n^{(k)}, h_n^{(k)} \in [0, 1], \sum_{k=1}^K h_n^{(k)} = 1, \quad p(\mathbf{f}^{(k)}) = \mathcal{N}(\mathbf{f}^{(k)}|\mathbf{m}_k, \Sigma_k) \quad (1)$$

where $\Sigma_k = [\mathcal{K}(\mathbf{x}_n, \mathbf{x}_m)]$ is a covariance matrix and $\mathcal{K}(\cdot, \cdot)$ is a kernel function, and we set $\mathbf{m}_k = \mathbf{0}$ with out losing generality. From the Bayesian perspective, this is equivalent to placing a Dirac delta prior over $\pi_{nk} : \pi_{nk} \sim \delta(\pi_{nk} - h_n^{(k)})$, where the inductive bias is encoded by the prior distribution. By introducing $h^{(k)}(\mathbf{f}_n)$, we essentially convert a multi-label problem into a multi-class problem as $\boldsymbol{\pi}_n$ encodes the probability of assigning $\mathbf{x}_n$ to each of the $K$ components. Specifically, given the learned mixture components $\boldsymbol{\theta}_k$'s, for a test data sample $\mathbf{x}_*$, we predict the component assignments $\boldsymbol{\pi}_*$ using the trained GP. The final labels are obtained as $p(\mathbf{y}_*|\Theta) = \sum_{k=1}^K \pi_{*k}\boldsymbol{\theta}_k$.

Posterior inference of latent variables in the two phases are jointly performed by maximizing the log marginal likelihood of the observed multiple labels for all training samples:

$$\ln p(\mathbf{Y}|\mathbf{X}) = \ln \int \int \sum_Z \prod_n \prod_l \prod_k p(\mathbf{f}^{(k)})p(\theta_{kl})p(z_{nk}|\mathbf{f}_n)p(y_{nl}|z_{nk}, \theta_{kl})\mathrm{d}F\mathrm{d}\Theta \quad (2)$$

Directly maximizing this likelihood is intractable due to the interplay of the latent variables. So we turn to optimizing the evidence lower bound (ELBO) of the log marginal: $\mathcal{L}(q) = \int q(\Theta, Z, F) \ln \frac{p(\mathbf{Y}, Z, F, \Theta|\mathbf{X})}{q(\Theta, Z, F)} \mathrm{d}\Theta \mathrm{d}Z \mathrm{d}F$, where $q(\Theta, Z, F)$ is the variational distribution. However, a key challenge that prevents us from using the standard mean field variational inference (MF-VI) is the term $p(z_{nk}|\mathbf{f}_n)$, defined by the mapping function $h_n^{(k)}$ in (1). As the most typical forms of $h_n^{(k)}$ (e.g., softmax) are non-conjugate with the prior distribution $p(\mathbf{f}^{(k)})$, which is a Gaussian, the variational posterior $q(\mathbf{f}^{(k)})$ cannot be derived analytically.

## 3.2 Auxiliary Variables based Variational Inference

We propose to resolve the non-conjugate mapping function in the complete data likelihood by introducing a number of auxiliary latent variables such that the augmented complete data likelihood becomes conjugate. Auxiliary variables have been used in MCMC based inference, such as slice sampling [22] and Hamiltonian MCMC [23], with improved sampling efficiency. The basic idea of auxiliary variables based variational inference (AV-VI) is to apply the following transformation: $p(x) = \int_y p(x|y)p(y)\mathrm{d}y$, where $p(x)$ is a target function that is difficult to compute (e.g., non-conjugate) during VI. If the conditional likelihood $p(x|y)$ is still non-conjugate, this process will continue until a conjugate conditional is achieved.

A key identity that we leverage to achieve a conditional likelihood conjugate to a Gaussian prior $p(\mathbf{f}^{(k)})$ is to convert a logistic sigmoid function as a scale mixture of Gaussian's [24] where the mixture is defined by a Pólya-Gamma distribution $p(\omega) = \mathrm{PG}(\omega|b, 0)$,

$$\frac{(e^f)^a}{(1 + e^f)^b} = 2^{-b} e^{\kappa f} \int_0^\infty e^{\frac{-\omega f^2}{2}} p(\omega)\mathrm{d}\omega \tag{3}$$

where $b \geq 0, \kappa = a - \frac{b}{2}$. However, a sigmoid function is only suitable for binary classification, making it infeasible for a mapping function that outputs the assignments for $K > 2$ components. Thus, we adopt the logistic-softmax function [25] as our mapping function:

$$h_n^{(k)} = p(z_{nk} = 1|\mathbf{f}_n) = \frac{\sigma(f_n^{(k)})}{\sum_{j=1}^K \sigma(f_n^{(j)})} \tag{4}$$

To handle the summation in (4), we introduce random variables $\boldsymbol{\lambda}_{1:N}$ and use identity $\frac{1}{x} = \int_0^\infty e^{-\lambda x}\mathrm{d}\lambda$ so that

$$p(z_{nk} = 1|\mathbf{f}_n, \lambda_n) = \sigma(f_n^{(k)}) \prod_{j=1}^K e^{-\lambda_n \sigma(f_n^{(j)})} \tag{5}$$

where $p(\lambda_n) \propto \mathbb{1}_{(0,\infty)}, \forall n \in [1, N]$. By leveraging the moment generation function of the Poisson distribution $\mathrm{Po}(\lambda)$, we introduce random variables $\Upsilon = \{\boldsymbol{\upsilon}_1, ..., \boldsymbol{\upsilon}_N\}$, where $\boldsymbol{\upsilon}_n = (\upsilon_{n1}, ..., \upsilon_{nK})^T$, to convert the exponential term in (5), which leads to

$$p(z_{nk} = 1|\mathbf{f}_n, \boldsymbol{\upsilon}_n) = \sigma(f_n^{(k)}) \prod_{j=1}^K (\sigma(-f_n^{(j)}))^{\upsilon_{nj}} \tag{6}$$

where $\upsilon_{nk} \sim \mathrm{Po}(\upsilon_{nk}|\lambda_n)$. Finally, using (3) and introducing the Pólya-Gamma random variables $\Omega = \{\boldsymbol{\omega}_1, ..., \boldsymbol{\omega}_N\}$, where $\boldsymbol{\omega}_n = (\omega_{n1}, ..., \omega_{nK})^T$, leads to

$$p(z_{nk} = 1|\mathbf{f}_n, \upsilon_{nk}, \omega_{nk}) = \prod_{k=1}^K 2^{-(z_{nk} + \upsilon_{nk})} \exp\left\{\frac{(z_{nk} - \upsilon_{nk})f_n^{(k)}}{2} - \frac{(f_n^{(k)})^2}{2}\omega_{nk}\right\} \tag{7}$$

where $\omega_{nk} \sim \mathrm{PG}(\omega_{nk}|\upsilon_{nk}, 0)$. Figure 3 shows the graphical model with the auxiliary variables ($\mathbf{x}_n$'s are omitted from the graph to keep the notation uncluttered).

We proceed by defining a variational distribution with auxiliary variables:

$$q(\Theta, Z, F, \boldsymbol{\lambda}, \Upsilon, \Omega) = q(\Theta)q(Z)q(F)q(\boldsymbol{\lambda})q(\Upsilon, \Omega) \tag{8}$$

The optimal variational distribution can be obtained by computing the moments of component variational distributions using some important properties of the main and auxiliary variables and iterating until convergence. The optimal variational distributions of the main latent variables are summarized in the following theorem.

**Theorem 1.** *With the auxiliary random variables and the transformed complete conditional likelihood given in (6), the optimal components of the variational distribution as specified by (8) are given by*

- *Component assignments $\widehat{q}(Z) = \prod_n \prod_k \widehat{q}(z_{nk})$:*

$$\widehat{q}(z_{nk}) = Cat(z_{nk}|\widehat{\phi_{nk}}); \quad \widehat{\phi_{nk}} \propto \exp\left\{\sum_{l=1}^L [y_{nl}(\psi(\widehat{a_{kl}}) - \psi(\widehat{a_{kl}} + \widehat{b_{kl}}))] + \frac{\widehat{m_{nk}}}{2}\right\} \tag{9}$$

*where $\psi(\cdot)$ is the digamma function and $\widehat{m_{nk}}$ is $n$-th element of mean of $\widehat{q}(\mathbf{f}^{(k)})$ defined in $\widehat{q}(F)$.*

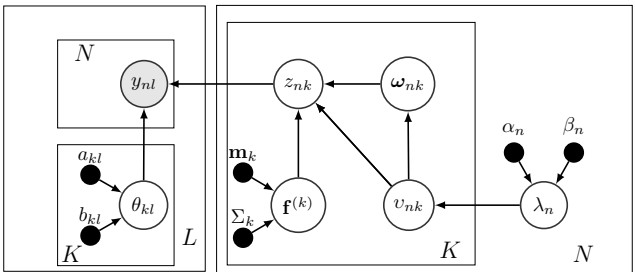

Figure 3: Graphical model with auxiliary variables

- *Bernoulli mixture components $\widehat{q}(\Theta) = \prod_k \prod_l \widehat{q}(\theta_{kl})$*

$$\widehat{q}(\theta_{kl}) = Beta(\theta_{kl}|\widehat{a_{kl}}, \widehat{b_{kl}}); \quad \widehat{a_{kl}} = a_{kl} + \sum_{n=1}^{N} \widehat{\phi_{nk}} y_{nl}, \widehat{b_{kl}} = b_{kl} + \sum_{n=1}^{N} \widehat{\phi_{nk}}(1 - y_{nl}) \quad (10)$$

- *GP latent functions $\widehat{q}(F) = \prod_k \widehat{q}(\mathbf{f}^{(k)})$:*

$$\widehat{q}(\mathbf{f}^{(k)}) = \mathcal{N}(\mathbf{f}_k|\widehat{\mathbf{m}_k}, \widehat{\Sigma_k}); \widehat{\mathbf{m}_k} = \frac{1}{2}\widehat{\Sigma_k}(\widehat{\phi_k} - \mathbb{E}_{\widehat{q}(\boldsymbol{v}_k)}[\boldsymbol{v}_k]), \widehat{\Sigma_k} = (\Sigma_k^{-1} + diag(\mathbb{E}_{\widehat{q}(\boldsymbol{\omega}_k, \boldsymbol{v}_k)}[\boldsymbol{\omega}_k]))^{-1}$$
(11)
*where $\boldsymbol{v}_k = (v_{1k}, ..., v_{Nk})^T$, $\boldsymbol{\omega}_k = (\omega_{1k}, ..., \omega_{Nk})^T$, and $\widehat{q}(\boldsymbol{\omega}_k, \boldsymbol{v}_k) = \widehat{q}(\boldsymbol{\omega}_k|\boldsymbol{v}_k)\widehat{q}(\boldsymbol{v}_k)$ is the optimal variational distribution for these auxiliary variables.*

The specific forms of the auxiliary variational distributions $q(\boldsymbol{\lambda})$ and $q(\Upsilon, \Omega)$ are provided in Appendix B as part of the detailed proof of the theorem.

**Model interpretation.** The optimal variational distributions are fairly intuitive. Interpreting these distributions can reveal some key insights on how the proposed GP-B$^2$M model leverages the data features as an effective inductive bias to discover semantically coherent components from a sparse label space. First, from (9), the component assignment of data sample $(\mathbf{x}_n, \mathbf{y}_n)$ is determined by two terms: the first term indicates how all its labels $\mathbf{y}_n$ are correlated with the component and the second term reflects how likely to categorize the features $\mathbf{x}_n$ into to the component. Second, from (10), since the component assignments are further utilized to compute the Bernoulli mixture components, the optimal components naturally aggregate both label and feature information to ensure semantic consistency as a result of using data features as the inductive bias. Last, from (11), the GP latent function value on a component increases with a positive component assignment and decreases with a 'negative' assignment, captured by the Poisson auxiliary variables $\boldsymbol{v}_k$.

**Time complexity.** According to (11), posterior inference of GP-B$^2$M has the computational complexity of $O(N^3K)$ which is identical to training $K$ GPs. Since each component can be updated independently, we can parallelize the computation to further reduce the complexity to $O(N^3)$. We can further leverage sparse kernel machines (e.g., Sparse GP) to reduce the complexity if N is large.

### 3.3 Multi-Label Active Sampling

Being a Bayesian model, GP-B$^2$M outputs the predictive distribution that provides both the label prediction and a label covariance matrix. As the covariance matrix captures both the uncertainty of individual labels and correlation of each pair of labels , it provides essential information to design a principled measure to quantify a data sample's overall contribution to a correlated label space.

For each testing sample $\mathbf{x}_*$, the predictive mean can be computed using the variational distributions:

$$\mathbb{E}[\mathbf{y}_*|\mathbf{x}_*] \approx \sum_k \mathbb{E}_{p(\mathbf{f}_*|\mathbf{X}, \mathbf{Y}, \mathbf{x}_*)}[\pi_{*k}]\mathbb{E}_{q(\Theta)}[\boldsymbol{\theta}_k]$$

$$p(\mathbf{f}_*|\mathbf{X}, \mathbf{Y}, \mathbf{x}_*) \approx \int p(\mathbf{f}_*|\mathbf{X}, F, \mathbf{x}_*)q(F)\mathrm{d}F, \quad \mathbb{E}_{q(\boldsymbol{\theta}_k)}[\theta_{kl}] = \widehat{a_{kl}}/(\widehat{a_{kl}} + \widehat{b_{kl}})$$
(12)

where $\pi_{*k} = p(z_{*k} = 1|\mathbf{f}_*)$ is defined as a logistic-softmax function given in (4). Theorem 1 shows that $q(F)$, which approximates the true posterior $p(F|\mathbf{X}, \mathbf{Y})$, is a Gaussian. Hence, $p(\mathbf{f}_*|\mathbf{X}, \mathbf{Y}, \mathbf{x}_*)$ is also a Gaussian. However, the logistic-softmax transformation makes predictive mean intractable

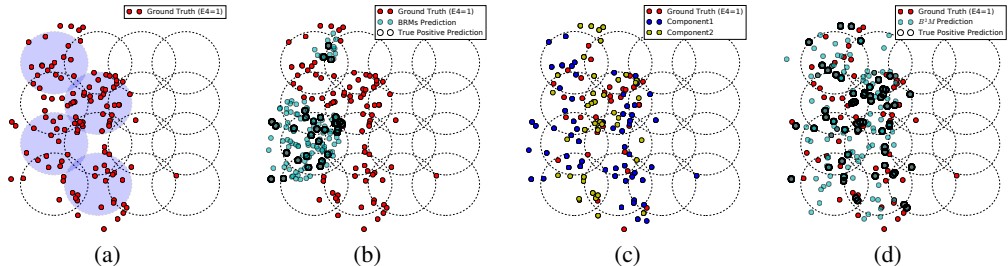

Figure 4: (a) Distribution of $E4$ samples; (b) Prediction by BRMs; (c) Mixture component assignments by B$^2$M; (d) Prediction by B$^2$M.

to compute. We propose to conduct Monte Carlo (MC) integration by drawing samples from $p(\mathbf{f}_*|\mathbf{X}, \mathbf{Y}, \mathbf{x}_*)$, perform logistic-softmax transformation, and then average.

The GP-B$^2$M model also allows us to compute the predicted label covariance,

$$\text{cov}[\mathbf{y}_*|\mathbf{x}_*] = \sum_k \mathbb{E}[\pi_{*k}]\{\mathbb{E}[\Lambda_k] + \mathbb{E}[\boldsymbol{\theta}_k]\mathbb{E}[\boldsymbol{\theta}_k]^T\} - \mathbb{E}[\mathbf{y}_*|\mathbf{x}_*]\mathbb{E}[\mathbf{y}_*|\mathbf{x}_*]^T \tag{13}$$

where $\Lambda_k = \text{diag}\{\mathbb{E}[\theta_{kl}](1 - \mathbb{E}[\theta_{kl}])\}$. The predicted label covariance captures both individual label uncertainty (diagonal entries of the matrix) and label correlations (off-diagonal entries), which is instrumental to quantify the total uncertainty of a test sample with respect to its predicted labels. Since directly computing the entropy of a mixture distribution is challenging, we instead choose to use the log determinant of covariance matrix: $\ln|\text{cov}[\mathbf{y}_*|\mathbf{x}_*]|$, as a proxy for uncertainty evaluation. Intuitively, this is equivalent to approximating $p(\mathbf{y}_*|\mathbf{x}_*)$ using a multivariate Gaussian, whose entropy is the log determinant of its covariance matrix plus a constant.

The label covariance is computed using a point estimate of $\boldsymbol{\pi}_* = (\pi_{*1}, ..., \pi_{*K})^T$ (one $\pi_{*k}$ for each class) to quantify the total uncertainty on the label side. As a Bayesian model, the proposed GP-B$^2$M allows us to quantify the variation of each $\pi_k$ using its predictive variance. Through MC integration as described above, we compute the predictive variance $\text{Var}[\pi_{*k}]$ of sample $\mathbf{x}_*$ for each of the $K$ class. According to the property of the GP posterior, we can easily show that the model would assign a low variance to data samples near to the training data and a high variance to faraway samples. As a result, the predictive variance effectively captures the *feature uncertainty* that complements the label covariance. It allows the proposed sampling function to differentiate data samples based on their distinct contributions to model training and sample them accordingly. Our final **sampling function** is given by: $\hat{\mathbf{x}}_* = \arg\max_{\mathbf{x}_*} \ln|\text{cov}[\mathbf{y}_*|\mathbf{x}_*]| + \eta \sum_k \text{Var}[\pi_{*k}]/K$, where $\eta$ is used to balance between label covariance and predictive variance of data features. It can be dynamically updated to give a higher weight in the early stage of AL to the feature variance term for better exploration of the data space and then shift the focus to the label covariance term for effective fine-tuning of decision boundaries with a correct shape obtained through effective exploration.

## 4 Experiments

We conduct extensive experiments on both synthetic and real-world multi-label data to demonstrate: (1) important properties of GP-B$^2$M to capture complex label correlations and how they contribute to predict complex labels, (2) state-of-the-art ML-AL performance by comparing with existing competitive models, (3) impact of key model parameters through an ablation study, and (4) effectiveness of active sampling by examining sampled data instances.

### 4.1 Synthetic Data

We design a synthetic dataset with 18 labels that exhibit 4 distinct types of dependencies as defined in Figure 1 (c). In the introduction, we show that three discovered mixture components precisely capture some rather complex label dependencies (e.g., hierarchical and exclusive) while being highly interpretable. For this dataset, the model discovers 10 components in total and we show some other components in Appendix D along with their interpretations. We further demonstrate how the discovered components contribute to the prediction of more complex and less frequent labels. We use $E4$ as an example, which is located deep in the hierarchy and appears much less than other labels.

Figure 4 (a) shows the distribution of data samples whose labels contain $E4$. It can be seen that these samples are distributed across the entire $E1$ region (roughly corresponds to the shaded area in purple).

Note that, in addition to $E1$, $E4$ also depends on the exclusive relationship: $E3$ AND NOT $E2$ ($E2, E3$ are not shown in the figure to keep the distribution of $E4$ clear). Figure 4 (b) shows the prediction result from BRMs, which has very high false positive and negative rates. The poor performance is also reflected by a low ROC-AUC (area under the receiver operating characteristic curve) score at 0.58 (slightly better than random guessing). It appears that BRMs only predict correctly samples in an area where $E4$ samples are relatively dense while missing most others. This is because BRMs try to directly learn the feature-label mapping (by training independent binary predictors), which is usually weak for complex and less frequent labels, like $E4$.

Different from BRMs, the proposed GP-B$^2$M learns mixture components that correctly capture the label correlations and the final labels can be recovered by combining the mixture components through their predicted coefficients. As discussed earlier, $E4$ has a high chance to appear in either

Figure 5: AUC on different types of labels

| Label | G1-G8 | C1,C2 | O1-O4 | E1-E4 |
|---|---|---|---|---|
| BRMs | 0.79 | 0.68 | 0.72 | 0.58 |
| GP-B$^2$M | 0.83 | 0.70 | 0.86 | 0.82 |

Component 1 or 2. Figure 4 (c) shows the predicted component assignments and the top component is highlighted. As can be seen, most $E4$ samples are assigned Component 1 or 2 as their top component. Also, samples assigned to Component 1 (shown in blue) are mostly distributed in the non-overlapping geometric regions and those assigned to Component 2 (shown in yellow) are mostly in overlapping regions. By leveraging these components, GP-B$^2$M achieves much better prediction results as shown Figure 4 (d). Table 5 summarizes the AUC scores from BRMs and GP-B$^2$M on different types of labels. While both models achieve similar performance on some common labels (e.g., G), GP-B$^2$M significantly outperforms BRMs for more complex labels, where it is essential to capture important label correlations.

## 4.2 Real Data

**Datasets and experiment settings.** We choose five representative real-world multi-label datasets, including Delicious, Enron, Bibtex, Corel5K, and NUS-WIDE, from different application domains [26]. All datasets have a relatively large label space and high label sparsity ($2-6\%$). Table 2 in Appendix D summarizes key properties of the pre-processed datasets. We randomly shuffle each dataset and partition them into three parts: training, testing, and candidate pool. We keep a minimum of one positive instance per label in the initial training partition as required by BRMs based AL models. To make sure each label is well represented in each partition, we remove extremely rare labels with label frequency less than 20. Since the remaining labels are still highly imbalanced, we use the ROC-AUC score to evaluate the model performance. All the baseline models share the same copy of the initial training set to make a fair comparison. Active learning stops after each model selects 500 samples.

**Performance comparison.** We include five competitive baselines for AL performance comparison:

- **MMC** samples instances that introduce the greatest change of the expected loss. During label prediction, it uses logistic regression to predict the number of labels for a new instance [14].
- **Adaptive** considers both the separation margin of an SVM and the label cardinality inconsistency and combines these two parts for data sampling [16].
- **AUDI** uses a label ranking mechanism, where a dummy label is used to separate the positive and negative labels. Its sampling function is based on a modified cardinality inconsistency measure [27].
- **CVIRS** combines the ranking on the magnitude of the difference margin in predictions and the label vector inconsistency for active sampling [17].
- **CS-GP** conducts active sampling in a compressed label space using a multi-output GP [21].

Figure 6 reports the AUC scores of all the models. For each curve, we present the average result along with the error bar from 3 trials of randomly initialized AL experiments. The proposed B$^2$M model achieves better AL performance consistently on all the datasets. For multiple datasets, GP-B$^2$M establishes a clear advantage in the early to middle stages of AL. While a few baselines eventually converge to a similar AUC score, they usually take more iterations (by consuming more labels) to reach a comparable performance as GP-B$^2$M. In addition, by comparing with random sampling with the proposed sampling function, we clearly demonstrate that the superior AL performance attributes to both the Bayesian mixture model and the effective active sampling. Note that the AUDI model runs much slower for Bibtex and Corel5K when both the number of features and candidate pool size become very large so we omit the results.

**Analysis on label-wise improvement.** To further justify why the proposed GP-B$^2$M outperforms the baselines, we offer a fine-grained analysis on the label-wise performance, which helps to achieve

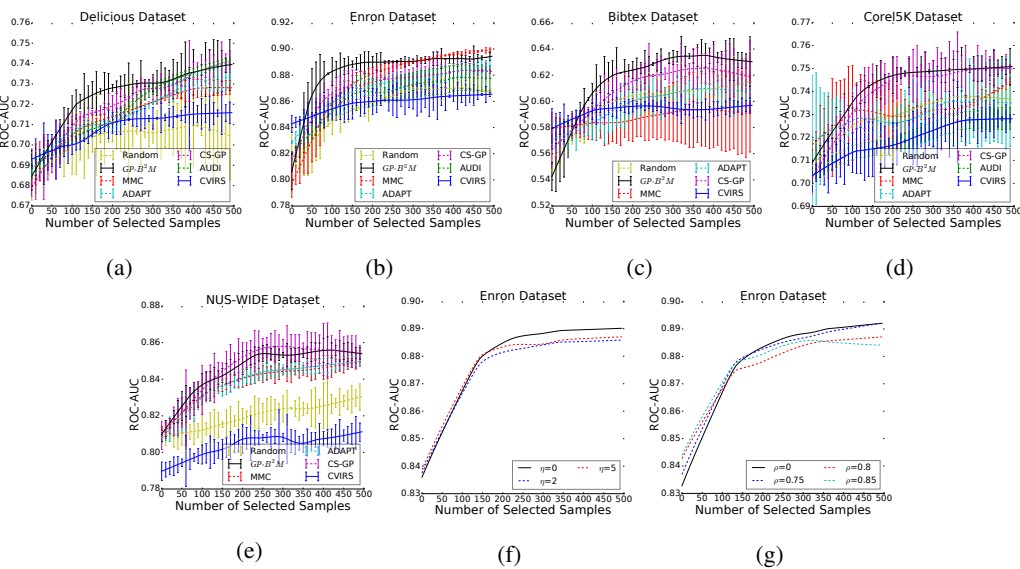

Figure 6: AL performance comparison (a)-(e); Ablation study on one example dataset (Enron): (f) shows the impact of $\eta$ and (g) shows the impact of effective components.

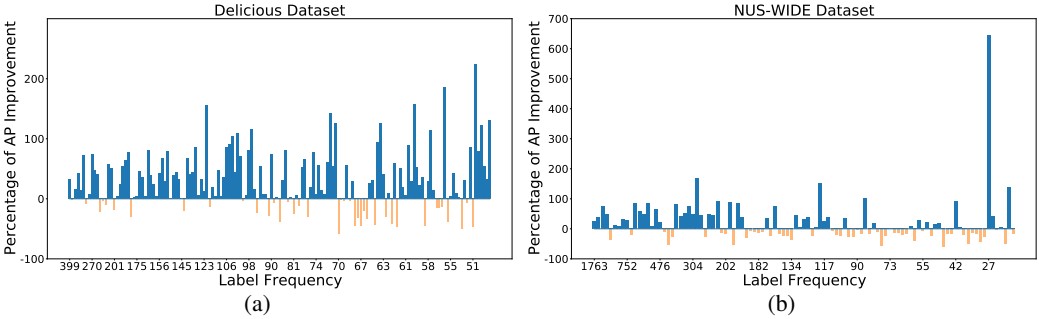

Figure 7: The label-wise average precision improvement over BRMs models.

a deeper insight on the performance gain. In fact, we reach a similar conclusion as the synthetic data experiments: while all models perform reasonably well on the most frequent labels, GP-B$^2$M clearly outperforms other baselines on less frequent labels, where considering the correlations with other labels can help to improve their predictions. Thus, in our fine-grained analysis, we first exclude the few most frequent labels and compute the Average Precision Improvement (API) of the $i$-th label,

$$API^{(i)} = \frac{AP_{B^2M}^{(i)} - AP_{BRMs}^{(i)}}{AP_{BRMs}^{(i)}} \times 100\% \tag{14}$$

where $AP_{B^2M}^{(i)}$ and $AP_{BRMs}^{(i)}$ denote the average precision of label $i$ provided by GP-B$^2$M and BRMs, respectively. Figure 7 demonstrates the label-wise $API$ on the NUS-WIDE and Delicious datasets as two illustrative examples. We observe that GP-B$^2$M performs significantly better than BRMs on labels with a moderate or low frequency. This is because those labels usually do not have sufficient positive instances for BRMs to learn independently. Furthermore, they may have a complex correlation with other labels. GP-B$^2$M effectively leverages label correlations to make better predictions on these labels. This further confirms the overall good performance of GP-B$^2$M.

**Ablation study.** We further investigate the impact of two tunable parameters of the model: (1) $\eta$, which balances label covariance and feature uncertainty for data sampling and (2) $\rho$, which controls the effective number of mixture components. Limited by space, we use the Enron dataset as an example and report other results in Appendix D. Figure 6 (f) compares the performance under different $\eta$ values. In early iterations, the label covariance guided sampling ($\eta$ is small) slightly falls behind the feature uncertainty guided sampling ($\eta$ is large) as the latter is more useful to explore

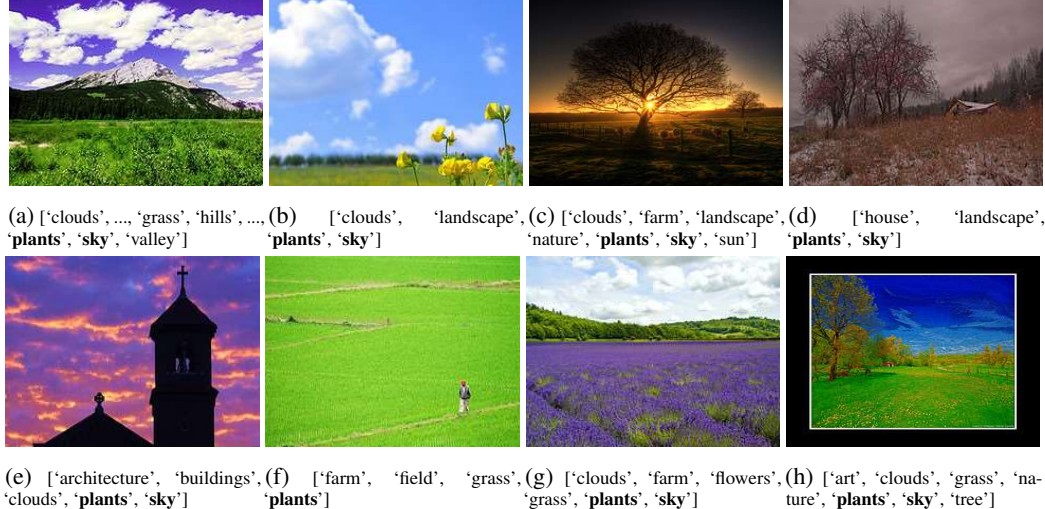

(a) ['clouds', ..., 'grass', 'hills', ..., **'plants'**, **'sky'**, 'valley'] (b) ['clouds', 'landscape', **'plants'**, **'sky'**] (c) ['clouds', 'farm', 'landscape', 'nature', **'plants'**, **'sky'**, 'sun'] (d) ['house', 'landscape', **'plants'**, **'sky'**]

(e) ['architecture', 'buildings', 'clouds', **'plants'**, **'sky'**] (f) ['farm', 'field', 'grass', **'plants'**] (g) ['clouds', 'farm', 'flowers', 'grass', **'plants'**, **'sky'**] (h) ['art', 'clouds', 'grass', 'nature', **'plants'**, **'sky'**, 'tree']

Figure 8: (a)-(c) Training images; (d)-(f) Images with a large feature uncertainty; (g)-(h) Images with a high label variance.

the feature space. Label covariance guided sampling gradually catches up and finally surpasses the feature uncertainty guided sampling. As shown next by the sampled instances, both criteria select informative instances that play complementary roles to improve the AL model. Figure 6 (g) shows how the AL performance is affected by the effective number of mixture components that is automatically determined by an upper bound $K$ and component strength ratio $\rho$. For component $k$, we compute its total 'effective posterior observations' [28] $g(\boldsymbol{\theta}_k) = \sum_{l=1}^{L} \widehat{a_{kl}} + \widehat{b_{kl}}$ and define a threshold as $\bar{g} = (\rho/K) \sum g(\boldsymbol{\theta}_k)$. The effective components only include those with $g(\boldsymbol{\theta}_k) \geq \bar{g}$. When the size of the training set is still small (early stage in AL), fewer components (large $\rho$) yield better results than more components (small $\rho$) by avoiding over-fitting. When more training labels are acquired, a more flexible model can better explain the label correlations thus has better performance.

**Examples of actively sampled instances.** To demonstrate the effectiveness of the proposed sampling function, we show images sampled by GP-B$^2$M from the NUS-WIDE dataset. To explain the distinct nature of these sampled images and how they contribute to the model training, we also show some representative images from the initial training pool for comparison. These correspond to images in Figure 8 (a)-(c) with common labels: 'plants' and 'sky'. First, for sampled images with a large feature uncertainty, while 'plants' and/or 'sky' are predicted for those images, they look very different from the training images. In particular, although the labels of the image in Figure 8 (e) contain both 'plants' and 'sky', there are no visible plants. For the image in Figure 8 (f), the label 'person' is not present even though a person is visible in the image. These types of samples are significantly dissimilar to the initial training set, thus considered valuable to explore the feature space for effective sampling. Images in Figure 8 (g)-(h) are samples based on a high label covariance. These images look similar to the training examples but their corresponding labels are somewhat different. Sampling these images can further improve the prediction of these labels and correlations there of, such as 'plants', 'grass', and 'nature'. As these images bring in additional labels, they may also help the model discover more possible correlations, such as that between 'plants' and 'flowers' or 'tree'.

## 5   Conclusion

We present a novel Gaussian Process-Bayesian Bernoulli Mixture (GP-B$^2$M) model for cost-effective multi-label active learning. GP-B$^2$M extracts global patterns of label correlations by learning from both (limited) training labels and data features. The mixture components, which are accurately learned from end-to-end and fully conjugate posterior inference, are capable of encoding complex label correlations while being highly interpretable. A novel sampling function is designed by combining feature uncertainty and label covariance, both of which can be obtained from the predictive distribution of the GP-B$^2$M model. Experiments conducted on both synthetic and real data justify the important properties of the model and its state-of-the-art AL performance.

## Acknowledgements

This research was supported in part by an NSF IIS award IIS-1814450 and an ONR award N00014-18-1-2875. The views and conclusions contained in this paper are those of the authors and should not be interpreted as representing any funding agency. We would also like to thank the anonymous reviewers for their constructive comments.

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
