# Appendix

**Organization.** The Appendix provides additional details, including the proof of the theoretical results and more experimental results, to support the major technical contribution presented in the main paper. The Appendix is organized as follows. Appendix A summarizes the major notations used in the main paper. Appendix B provides the detailed proof of Theorem 1. Appendix C presents the pseudo code for the active sampling process using GP-B$^2$M. Finally, Appendix D provides additional experimental results, including more ablation study, passive learning performance, and active sampling time comparison. The link to the source code is provide in Appendix E.

## A    Major Notations and Definitions

We summarize the major notations used in the main paper. We categorize the notations into four major types: observed, latent, auxiliary, and hyperparameters, based on their roles in the model. Table 1 provides a detailed definition of each notation along with their assigned type.

Table 1: Summary of notations with definitions

| Notation | Definition | Type |
|---|---|---|
| $\mathbf{x}_n$ | Feature vector of the $n$-th data sample | Observed |
| $\mathbf{y}_n$ | Output labels of $\mathbf{x}_n$ | Observed |
| $\mathbf{f}^{(k)}$ | Latent GP functions of component $k$ | Latent |
| $\mathbf{m}_k, \widehat{\mathbf{m}_k}$ | Prior and posterior means of $\mathbf{f}_k$ | Hyperparameter |
| $\Sigma_k, \widehat{\Sigma_k}$ | Prior and posterior covariances of $\mathbf{f}_k$ | Hyperparameter |
| $z_{nk}$ | Latent indicator variable | Latent |
| $\widehat{\phi_{nk}}$ | Posterior mean of latent indicator $z_{nk}$ | Hyperparameter |
| $h^{(k)}$ | Mapping function for component $k$ | Latent |
| $\boldsymbol{\pi}_n$ | Mixture component weights of $\mathbf{x}_n$ | Latent |
| $\boldsymbol{\theta}_k$ | Mixture component $k$ | Latent |
| $a_{kl}, b_{kl}; \widehat{a_{kl}}, \widehat{b_{kl}}$ | Prior and posterior parameters of Beta random variable $\theta_{kl}$ | Hyperparameter |
| $\lambda_n$ | Gamma auxiliary random variable | Auxiliary |
| $\alpha_n, \beta_n$ | Posterior parameter of Gamma random variable $\lambda_n$ | Hyperparameter |
| $\upsilon_{nk}$ | Poisson auxiliary random variable | Auxiliary |
| $\gamma_{nk}$ | Posterior mean of Poisson random variable $\upsilon_{nk}$ | Hyperparameter |
| $\omega_{nk}$ | Pólya-Gamma auxiliary variable | Auxiliary |
| $c_{nk}$ | Posterior parameter of Pólya-Gamma random variable $\omega_{nk}$ | Hyperparameter |

## B    Proof of Theorem 1

In this section, we provide the detailed proof of Theorem 1. We first prove the following lemma, which is a key component used in our proof.

**Lemma 1.** *Given the complete data likelihood* (7) *and the property of the Pólya-Gamma distribution* (15), *the posterior variational distribution* $\widehat{q}(\omega_{nk}, \upsilon_{nk})$ *of auxiliary variables* $\omega_{nk}$ *and* $\upsilon_{nk}$ *can be factorized as* $\widehat{q}(\omega_{nk}|\upsilon_{nk})\widehat{q}(\upsilon_{nk})$, *where* $\widehat{q}(\omega_{nk}|\upsilon_{nk})$ *is a Pólya-Gamma distribution and* $\widehat{q}(\upsilon_{nk})$ *is a Poisson distribution.*

*Proof.* We will make use of the property of the Pólya-Gamma distribution [24]. In particular, the probability density of a Pólya-Gamma distribution $\text{PG}(\omega|c_1, c_2)$ with parameters $c_1$ and $c_2$ can be derived through an exponential tilting of $\text{PG}(\omega|c_1, 0)$:

$$p(\omega|c_1, c_2) = \frac{\exp(-\frac{c_2^2}{2}\omega)p(\omega|c_1, 0)}{\mathbb{E}_{\omega'}[\exp(-\frac{c_2^2}{2}\omega')]} \tag{15}$$

where $p(\omega|c_1, 0)$ is the density of a $\text{PG}(\omega|c_1, 0)$ random variable and the expectation in the denominator is computed by:

$$\mathbb{E}_{\omega'}[\exp(-\omega't)] = \frac{1}{\cosh^{c_1}(\sqrt{t/2})} \tag{16}$$

where $\omega' \sim \text{PG}(c_1, 0)$.

By applying the general solution of mean field variational inference [28]

$$\ln \widehat{q}(\varphi_i) = \mathbb{E}_{\boldsymbol{\varphi}_{-i}}[\ln p(Y, \boldsymbol{\varphi})] + \text{Const} \tag{17}$$

where $\boldsymbol{\varphi} = \{\Theta, Z, F, \boldsymbol{\lambda}, \Upsilon, \Omega\}$ and $\varphi_i = \{\Upsilon, \Omega\}$, we have

$$\ln \widehat{q}(\Omega, \Upsilon) = \mathbb{E}_{Z,F,\boldsymbol{\lambda}}\left[\ln \prod_n \prod_k p(z_{nk}|f_n^{(k)}, \lambda_n, \upsilon_{nk}, \omega_{nk})p(f_n^{(k)})p(\omega_{nk}|\upsilon_{nk})p(\upsilon_{nk}|\lambda_n)\right] + \text{Const}$$

$$= \mathbb{E}_{Z,F,\boldsymbol{\lambda}}\left[\ln \prod_n \prod_k 2^{-(z_{nk}+\upsilon_{nk})} \exp\left\{\frac{(z_{nk} - \upsilon_{nk}f_n^{(k)})}{2} - \frac{(f_n^{(k)})^2}{2}\omega_{nk}\right\}. \tag{18}$$

$$\mathcal{N}(f_n^{(k)}|0, \Sigma_k(n, n))\text{PG}(\omega_{nk}|\upsilon_{nk}, 0)\frac{\lambda_n^{\upsilon_{nk}} \exp(-\lambda_n)}{\upsilon_{nk}!}\right] + \text{Const}$$

$$= \mathbb{E}_{Z,F,\boldsymbol{\lambda}}\left[\sum_n \sum_k \left\{-(z_{nk} + \upsilon_{nk})\ln 2 + \frac{z_{nk} - \upsilon_{nk}f_n^{(k)}}{2} - \frac{f_n^{(k)2}}{2}\omega_{nk}\right.\right. \tag{19}$$

$$\left.\left. + \text{PG}(\omega_{nk}|\upsilon_{nk}, 0) + \upsilon_{nk}\ln \lambda_n - \ln \upsilon_{nk}!\right\}\right] + \text{Const}$$

$$= \sum_n \sum_k \left\{-(\mathbb{E}_{z_{nk}}[z_{nk}] + \upsilon_{nk})\ln 2 - \upsilon_{nk}\frac{\mathbb{E}_{f_n^{(k)}}[f_n^{(k)}]}{2} - \omega_{nk}\frac{\mathbb{E}_{f_n^{(k)}}[f_n^{(k)2}]}{2}\right. \tag{20}$$

$$\left. + \ln \text{PG}(\omega_{nk}|\upsilon_{nk}, 0) + \upsilon_{nk}[\psi(\alpha_n) - \beta_n] - \ln \upsilon_{nk}!\right\} + \text{Const} \tag{21}$$

which implies that $q(\omega_{nk}, \upsilon_{nk})$ follows the distribution given below:

$$\widehat{q}(\omega_{nk}, \upsilon_{nk}) \propto \left(\exp(-\frac{\widehat{m_{nk}}}{2})\right)^{\upsilon_{nk}} \exp\left(-\frac{(\bar{f}_n^{(k)})^2}{2}\omega_{nk}\right)\text{PG}(\omega_{nk}|\upsilon_{nk}, 0)\left(\frac{\exp(\psi(\alpha_n))}{\beta_n}\right)^{\upsilon_{nk}}\frac{1}{\upsilon_{nk}!} \tag{22}$$

$$\propto \left\{\exp\left(-\frac{(\bar{f}_n^{(k)})^2}{2}\omega_{nk}\right)\text{PG}(\omega_{nk}|\upsilon_{nk}, 0)\cosh^{\upsilon_{nk}}\left(-\frac{\bar{f}_n^{(k)}}{2}\right)\right\}. \tag{23}$$

$$\left\{\frac{\exp(\psi(\alpha_n))\exp(\frac{\widehat{m_{nk}}}{2})}{\beta_n \cosh(-\frac{\bar{f}_n^{(k)}}{2})}\right\}^{\upsilon_{nk}}\frac{1}{\upsilon_{nk}!} \tag{24}$$

$$\propto \text{PG}(\omega_{nk}|\upsilon_{nk}, c_{nk})\text{Poisson}(\upsilon_{nk}|\gamma_{nk}) \tag{25}$$

where

$$c_{nk} = \bar{f}_n^{(k)} = \sqrt{\widehat{m_{nk}}^2 + \widehat{\Sigma_k}(n, n)} \tag{26}$$

$$\gamma_{nk} = \frac{\exp(\psi(\alpha_n))\exp(\frac{\widehat{m_{nk}}}{2})}{\beta_n \cosh(\frac{\bar{f}_n^{(k)}}{2})} \tag{27}$$

Here $\widehat{m_{nk}}$ denotes the $n^{th}$ element of $\widehat{\mathbf{m}}_k$, and $\widehat{\Sigma_k}(n, n)$ denotes the $n^{th}$ element on the diagonal of $\widehat{\Sigma}_k$. $\qquad\square$

**Proof of Theorem 1**

Now we provide the proof for Theorem 1 in the main paper by deriving the variational posterior distribution. We start by specifying the complete data likelihood

$$p(\mathbf{Y}, \boldsymbol{\varphi}) = \prod_n \prod_k p(y_{nk}|\boldsymbol{\theta}_k, z_{nk})p(\boldsymbol{\theta}_k)p(z_{nk}|\mathbf{f}_n, \lambda_n, \upsilon_{nk}, \omega_{nk})p(\omega_{nk}|\upsilon_{nk})p(\upsilon_{nk}|\lambda_{nk})p(\lambda_n) \tag{28}$$

Based on the definition of the variational distribution given by (8), we invoke the general solution in (17) iteratively by setting $\varphi_i$ as $\Theta, \boldsymbol{\lambda}, Z$, and $F$, respectively.

**Optimize with respect to $q(\Theta)$**

$$\ln \widehat{q}(\Theta) = \mathbb{E}_Z[\ln p(\mathbf{Y}|Z, \Theta)p(\Theta)] + \text{Const} \tag{29}$$

$$= \sum_{n=1}^{N}\sum_{k=1}^{K} \mathbb{E}[z_{nk}] \sum_{l=1}^{L} y_{nl} \ln \theta_{kl} + (1 - y_{kl})\ln(1 - \theta_{kl})$$

$$+ \sum_{k=1}^{K}\sum_{l=1}^{L}(a_{kl} - 1)\ln\theta_{kl} + (b_{kl} - 1)\ln(1 - \theta_{kl}) + \text{Const} \tag{30}$$

$$= \sum_{k=1}^{K}\sum_{l=1}^{L}\left[\sum_{n=1}^{N}(\widehat{\phi_{nk}}y_{nl} + a_{kl} - 1)\ln\theta_{kl} + ((\widehat{\phi_{nk}}(1 - y_{nl}) + b_{nk} - 1)\ln(1 - \theta_{kl})\right] + \text{Const} \tag{31}$$

which implies that $\widehat{q}(\boldsymbol{\theta}_k) \sim \prod_{k=1}^{K} \text{Beta}(\theta_{kl}|\widehat{a_{kl}}, \widehat{b_{kl}})$ where

$$\widehat{a_{kl}} = a_{kl} + \sum_{n=1}^{N} \widehat{\phi_{nk}}y_{nl} \tag{32}$$

$$\widehat{b_{kl}} = b_{kl} + \sum_{n=1}^{N} \widehat{\phi_{nk}}(1 - y_{nl}) \tag{33}$$

**Optimize with respect to $q(\boldsymbol{\lambda})$**

$$\ln \widehat{q}(\boldsymbol{\lambda}) = \mathbb{E}_\Upsilon[\ln p(\boldsymbol{\upsilon}|\boldsymbol{\lambda})p(\boldsymbol{\lambda})] + \text{Const} \tag{34}$$

$$= \sum_{n=1}^{N}\left[\sum_{k=1}^{K}\gamma_{nk}\ln\lambda_n - K\lambda_n\right] + \text{Const} \tag{35}$$

where computation of $\gamma_{nk}$ is given by (27) in Lemma 1. This implies that $\widehat{q}(\lambda_n) \sim \text{Gamma}(\lambda_n|\alpha_n, \beta_n)$ where

$$\alpha_n = \sum_{k=1}^{K}\gamma_{nk} + 1 \tag{36}$$

$$\beta_n = K \tag{37}$$

**Optimize with respect to $q(F)$**

$$\ln \widehat{q}(F) = \mathbb{E}_{Z,\Upsilon,\Omega}\left[\ln \prod_{n=1}^{N}\prod_{k=1}^{K} p(z_{nk}|f_n^{(k)}, \lambda_n, \upsilon_{nk}, \omega_{nk})p(f_n^{(k)})\right] \tag{38}$$

$$= \mathbb{E}_{Z,\Upsilon,\Omega}\left[\ln \prod_{k=1}^{K} \mathcal{N}(\mathbf{f}^{(k)}|\frac{\mathbf{z}_k - \boldsymbol{\upsilon}_k}{2}, \text{diag}(\boldsymbol{\omega}_k)^{-1})\mathcal{N}(\mathbf{f}^{(k)}|\mathbf{0}, \Sigma_k)\right] \tag{39}$$

$$\tag{40}$$

which implies that $\widehat{q}(\mathbf{f}^{(k)}) \sim \mathcal{N}(\mathbf{f}^{(k)}|\widehat{\mathbf{m}}_k, \widehat{\Sigma}_k)$ where

$$\widehat{\mathbf{m}}_k = \frac{1}{2}\widehat{\Sigma}_k(\widehat{\boldsymbol{\phi}_k} - \mathbb{E}[\boldsymbol{\upsilon}_k]) \tag{41}$$

$$\widehat{\Sigma}_k = (\Sigma_k^{-1} + \text{diag}(\mathbb{E}[\boldsymbol{\omega}_k]))^{-1} \tag{42}$$

where $\mathbb{E}[\boldsymbol{\upsilon}_k] = \boldsymbol{\gamma}_k$, $\mathbb{E}_{q(\omega_{nk}, \upsilon_{nk})}[\omega_{nk}] = \frac{\mathbb{E}[z_{nk}] + \gamma_{nk}}{2c_{nk}}\tanh\frac{c_{nk}}{2}$ [24].

**Optimize with respect to** $q(\mathbf{Z})$

$$\ln \widehat{q}(Z) = \mathbb{E}_{\Theta, F, \Omega, \Upsilon} \left[ \ln \prod_n \prod_k \prod_l p(y_{nl}|\boldsymbol{\theta}_k) p(z_{nk}|f_n^{(k)}, \omega_{nk}, \upsilon_{nk}) \right] + \text{Const} \tag{43}$$

$$= \mathbb{E}_{\Theta, F, \Omega, \Upsilon} \left[ \sum_n \sum_k z_{nk} \left( \sum_l \ln \mu_{kl} + (1 - y_{nl}) \ln(1 - \mu_{kl}) \right) \right.$$

$$\left. - (z_{nk} + \upsilon_{nk}) \ln 2 + \frac{(z_{nk} - \upsilon_{nk}) f_n^{(k)}}{2} - \frac{(f_n^{(k)})^2}{2} \omega_{nk} \right] + \text{Const} \tag{44}$$

which implies that $\widehat{q}(z_{nk}) \sim \text{Cat}(z_{nk}|\widehat{\phi_{nk}})$ where

$$\widehat{\phi_{nk}} \propto \exp \left\{ \sum_{l=1}^L [y_{nl}(\psi(\widehat{a_{kl}}) - \psi(\widehat{a_{kl}} + \widehat{b_{kl}}))] + \frac{\widehat{m_{nk}}}{2} \right\} \tag{45}$$

## C  Pseudo Code for GP-B²M Based Active Learning

---

**Algorithm 1:** Active sampling using GP-B²M

---
**input**  :Training set: $(\mathbf{X}, \mathbf{Y})$, kernel function: $\mathcal{K}(\cdot)$, unlabeled candidate pool: $\mathbf{X}_u$
**output** :Selected sample: $\hat{\mathbf{x}}_*$
1 Initialize variational hyperparameters: $\mathbf{m}_k, a_{kl}, b_{kl}, \alpha_n, \gamma_{nk}$
2 Set $\Sigma_k = \mathcal{K}(\mathbf{X}, \mathbf{X})$, $\beta_n = K$.
3 **while** *(!converged)* **do**
4    **for** $n \in [1:N]$ **do**
5       **for** $k \in [1:K]$ **do**
6          update $\gamma_{nk}$ using equation (27)
7          update $c_{nk}$ using equations (23) and (26)
8          update $\widehat{\phi_{nk}}$ using equation (45)
9       **end**
10   **end**
11   **for** $k \in [1:K]$ **do**
12       update $\widehat{\mathbf{m}}_k$ using equation (41)
13       update $\widehat{\Sigma}_k$ using equation (42)
14       **for** $l \in [1:L]$ **do**
15          update $\widehat{a_{kl}}$ using equation (32)
16          update $\widehat{b_{kl}}$ using equation (33)
17       **end**
18   **end**
19 **end**
20 return $\hat{\mathbf{x}}_* = \arg \max_{\mathbf{x}_* \in \mathbf{X}_u} \ln|\text{cov}[\mathbf{y}_*|\mathbf{x}_*]| + \eta \sum_k \text{Var}[\pi_{*k}]/K$

---

## D  More Details of Experiments and Additional Results

In this section, we first provide more details about our experiments, including the key properties of the real-world data (see Table 2) and hyperparameter settings. We then present some additional experimental results to complement the results in the main paper.

Our experiment runs on a High Performance Computing (HPC) cluster with Intel® Xeon® Gold 6150 CPUs @ 2.70GHz (six cores per learning task), 24 TB RAM, and 100 Gbit/sec RoCEv2 interconnect (Mellanox MLX5/Juniper QFX210-64c). The submitted source code does not require GPUs to run.

### D.1  Hyperparameter Settings

The parameters of the prior Beta distribution, $a_{kl}$ and $b_{kl}$ are set to 1. We determine the model convergence by observing the sum of squared changes of the latent random variables, $\Delta \varphi$, between

Table 2: Summary of datasets

| Dataset | Instances | Features | Labels | Sparsity |
|---------|-----------|----------|--------|----------|
| Delicious | 6833 | 500 | 156 | 0.04 |
| Enron | 1702 | 1001 | 53 | 0.06 |
| Bibtex | 7013 | 1836 | 127 | 0.02 |
| Corel5K | 5000 | 499 | 132 | 0.02 |
| NUS-WIDE | 269,648 | 64 | 128 | 0.02 |

two consecutive parameter update iterations. The parameter update ceases if $\Delta\varphi \leq 1e^{-3}$ or the number of update iterations exceeds 20. In the experiment we observe the $B^2M$ converges fast, often within 5 to 8 iterations.

Both MIML [20] and CS-GP [21] compress the original label space though compressed sensing. We adopt Bayesian principle component analysis to adjust the optimal compressing rate for CS-GP as proposed in [21]. We then apply the same compressing rate to MIML to make a fair comparison. On average, the compressing rates applied by both model is close to $0.45 \pm 0.05$ on the five datasets. We set $\rho = 0.75$ and start $B^2M$ training with $K = 25$ components. We observe that as active learning goes, $K$ will gradually drop close to 10 for all the datasets. However, CBM [6] performs poorly at such small number of components so we fix the $K$ to 25 when training the model.

We use a RBF kernel: $\mathcal{K}(\mathbf{x}_1, \mathbf{x}_2) = \exp\{-\frac{|\mathbf{x}_1 - \mathbf{x}_2|^2}{2\delta^2}\}$ for $B^2M$ and other baselines that utilize the kernel machine for prediction. The length scale parameter $\delta$ for CS-GP is optimized via likelihood maximization and for the rest models are fixed as 1. Although we assume the assignment of each data instance is noise free, in $B^2M$, we still add a small noise - like term $\epsilon = 1e^{-3}$ to the diagonal of the gram matrix to ensure that the covariance matrix $\Sigma_k$ is positive definite. Finally, we follow the convention and set the prior mean of the latent GP functions $\mathbf{m}_k = \mathbf{0}, \forall k \in [1, K]$.

## D.2 Additional Mixture Components

We present some additional mixture components learned from the synthetic data that complement the three presented in the introduction of the main paper. As shown in Figure 9, the first component allocates high probability mass on $G3$, $G4$, and $G8$, which implies that it focuses on the union of these three geometric regions. Meanwhile, it also has the highest mass on $G6$ compared with other components. Since $E1$ depends on $G6$, we have a high chance to observe $E1$ and other type $E$ labels in this component as well. The second component focuses on a similar geometric region but pays less attention to $G3$ and $G6$. As a result, it is less likely to observe type $E$ labels, as evidenced by a low mass on $E1$. The third component only focuses on the intersection region of $G2$ and $G3$, as evidenced by a high mass on $O2$. In addition, a data instance assigned to this component is expected to have only one or two labels as indicated by the high mass on $C1$.

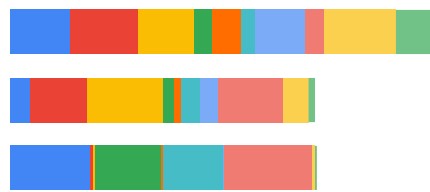

|G3|G4|G8|G2|G6|O2|O3|C1|C2|E1

Figure 9: Additional components from the synthetic data

## D.3 Passive Learning Performance

We report the passive learning performance of the proposed $B^2M$ along with some representative baselines, aiming to further demonstrate the effectiveness of the proposed active sampling function. As multiple baselines leverage BRMs as the base model with different sampling mechanisms, we only report the BRMs performance in the passive setting. In addition, we also include a compressed sensing based model (MIML) [20] and the conditional Bernoulli mixture model (CBM) [6]. As can be seen from Table 3, the passive learning performances of these two models are much lower than

other models, which indicates that they are less suitable when being trained using limited labeled data. As result, we did not include them for the active learning performance comparison in the main paper. From Table 3, we can also see that while $B^2M$ still outperforms other models in most cases under the passive setting, it achieves a more significant advantage in active learning. This further justifies the effectiveness of proposed active sampling function.

Table 3: Passive Learning Performance

| Dataset | Training size | $B^2M$ | CS-GP | MIML | CBM | BRMs |
|---------|---------------|--------|-------|------|-----|------|
| Corel5K | 100 | 0.71 | 0.70 | 0.61 | 0.64 | 0.70 |
| | 300 | 0.74 | 0.73 | 0.57 | 0.64 | 0.73 |
| | 500 | 0.74 | 0.75 | 0.60 | 0.65 | 0.73 |
| | 700 | 0.76 | 0.75 | 0.67 | 0.69 | 0.74 |
| BibTex | 100 | 0.58 | 0.58 | 0.59 | 0.50 | 0.57 |
| | 300 | 0.61 | 0.61 | 0.60 | 0.49 | 0.60 |
| | 500 | 0.61 | 0.61 | 0.62 | 0.53 | 0.62 |
| | 700 | 0.63 | 0.62 | 0.64 | 0.58 | 0.62 |
| NUS-WIDE | 100 | 0.81 | 0.82 | 0.66 | 0.66 | 0.80 |
| | 300 | 0.82 | 0.84 | 0.70 | 0.68 | 0.81 |
| | 500 | 0.83 | 0.85 | 0.70 | 0.69 | 0.82 |
| | 700 | 0.85 | 0.85 | 0.71 | 0.72 | 0.82 |
| Enron | 100 | 0.84 | 0.83 | 0.75 | 0.47 | 0.78 |
| | 300 | 0.86 | 0.84 | 0.69 | 0.49 | 0.80 |
| | 500 | 0.86 | 0.86 | 0.66 | 0.52 | 0.85 |
| | 700 | 0.87 | 0.88 | 0.61 | 0.55 | 0.86 |
| Delicious | 100 | 0.70 | 0.69 | 0.57 | 0.69 | 0.67 |
| | 300 | 0.70 | 0.71 | 0.60 | 0.70 | 0.68 |
| | 500 | 0.71 | 0.71 | 0.63 | 0.72 | 0.68 |
| | 700 | 0.75 | 0.72 | 0.65 | 0.74 | 0.69 |

## D.4 Additional Ablation Study Results

In addition to the results shown in the main paper, we present the remaining results that demonstrate the impact of the tunable parameters $\eta$ and $\rho$ over other real-world datasets. From Figure 10, we observe that the label covariance guided sampling usually leads to higher converged active learning performance while the variance guided sampling usually converges faster.

From Figure 11, we conclude that in general, the large number of components in the early stage of active learning might hurt the model performance due to the lack of training data and limited observations of label correlations. However, as active learning goes, the model needs more components to encode newly observed label correlations so that a larger $K$ usually leads to a better performance.

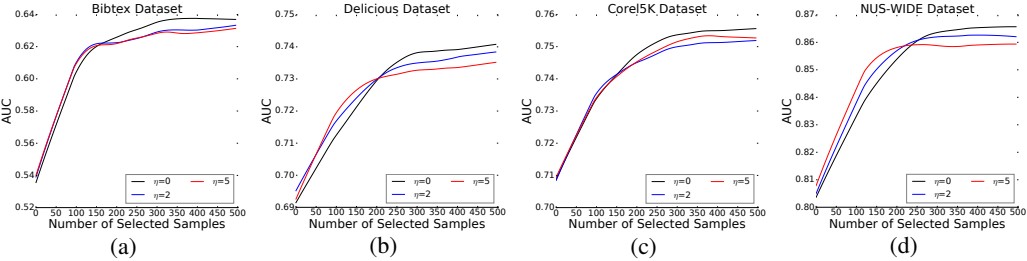

Figure 10: Impact of $\eta$

## D.5 Active Sampling Time Comparison

In Table 4, we present the execution time (in seconds) for a complete active sampling iteration that includes the model training and choosing the best sample from the unlabeled pool. The sample selection time for all the models is similar to each other since the prediction over candidate datasets

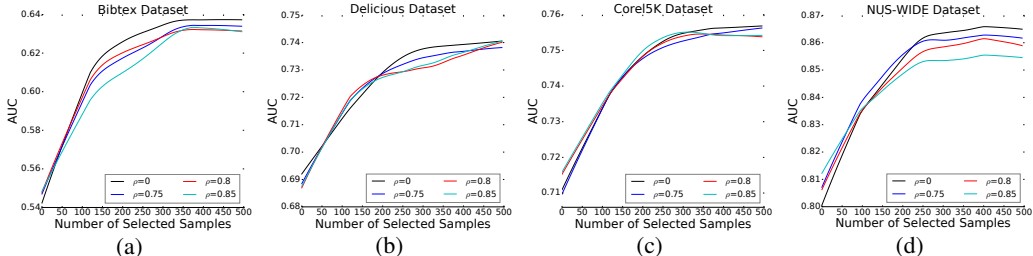

Figure 11: Impact of $\rho$

and the computation of the sampling criteria are linear. The major factor that affects the active sampling time is caused by the model retraining. This can be further decomposed to the total number of basic predictors multiplied by the time for training each basic predictor. Both B$^2$M and CBM have the least number of basic predictors to train ($K \leq 25$) thus run faster than other baselines. Both B$^2$M and CS-GP have the same complexity for training the basic predictors (i.e., GP). However, CS-GP runs slower as its optimal compressing rate (around 0.45) results in training more basic predictors. MMC and CVIRS leverage SVMs as the basic predictor which has the same learning complexity as GPs. Meanwhile, they need to train much more basic predictors than other baselines, making them the slowest models for active sampling. Finally, the MIML model relies on mutual information for sampling so it takes longer time than B$^2$M in most cases.

Table 4: Active Sampling Time

| Dataset | Training size | B$^2$M | CS-GP | MMC | CBM | CVIRS | MIML |
|---------|---------------|--------|-------|-----|-----|-------|------|
| Corel5K | Init | 10.7 | 19.1 | 17.0 | 2.9 | 19.0 | 16.8 |
| | 100 | 12.1 | 22.8 | 37.1 | 3.6 | 34.8 | 16.8 |
| | 300 | 17.4 | 32.5 | 96.4 | 9.5 | 78.3 | 17.1 |
| | 500 | 23.7 | 42.3 | 157.5 | 16.5 | 149.4 | 17.0 |
| BibTex | Init | 24.9 | 37.7 | 34.5 | 7.9 | 38.4 | 19.4 |
| | 100 | 26.1 | 47.6 | 62.5 | 9.6 | 61.6 | 44.1 |
| | 300 | 31.7 | 69.2 | 131.6 | 18.5 | 131.9 | 44.4 |
| | 500 | 38.1 | 89.9 | 201.7 | 27.3 | 232.7 | 44.2 |
| NUS-WIDE | Init | 2.5 | 2.3 | 3.5 | 0.4 | 11.2 | 8.6 |
| | 100 | 2.8 | 3.6 | 6.0 | 0.5 | 20.7 | 8.7 |
| | 300 | 6.4 | 14.4 | 18.5 | 1.7 | 40.5 | 8.6 |
| | 500 | 10.9 | 21.3 | 51.8 | 4.9 | 62.7 | 8.5 |
| Enron | Init | 2.1 | 3.8 | 3.2 | 0.5 | 2.5 | 4.7 |
| | 100 | 3.9 | 4.8 | 18.9 | 3.8 | 8.7 | 4.8 |
| | 300 | 4.8 | 7.9 | 65.8 | 5.6 | 28.0 | 4.8 |
| | 500 | 8.4 | 18.6 | 122.8 | 11.3 | 56.4 | 5.0 |
| Delicious | Init | 11.1 | 15.5 | 20.1 | 4.7 | 52.9 | 31.9 |
| | 100 | 11.8 | 17.2 | 37.6 | 5.6 | 78.8 | 32.1 |
| | 300 | 16.6 | 23.4 | 92.5 | 8.8 | 143.6 | 31.8 |
| | 500 | 22.6 | 36.6 | 164.3 | 14.1 | 221.0 | 31.5 |

# E  Source Code

The source code and detailed documentation can be found at `https://github.com/ritmininglab/GP-B2M-MLAL.git`.