# OpenReview forum: "A Gaussian Process-Bayesian Bernoulli Mixture Model for Multi-Label Active Learning"
_NeurIPS.cc/2021/Conference — NeurIPS 2021 Poster_

### Official Review · Reviewer_Fw2q · 2021-07-17

**Rating:** 5
**Confidence:** 4

**Summary:**

Multi-label learning is an important yet challenging task in machine learning, particularly in extreme settings, where the label space is very large.  Despite that, quantifying the prediction uncertainty in multi-label learning for active learning is even harder, as labels can be correlated.  The paper presents a mixture model for multi-label classification that brings together the Gaussian process and Bernoulli mixture models,  which can be one of its contributions. The other contributions include the variation inference based on variable augmentation, and the use of the label covariance to compute the predictive uncertainty in the acquisition function.

**Limitations And Societal Impact:**

No, the authors did not address the limitations and potential negative social impact, which could be discussed from a perspective of applying the proposed active learning in practice.

**Main Review:**

* The proposed multi-label learning model is basically a simple mixture model, where each latent component is models as a distribution over all the possible labels, and each document is associated with a distribution over those latent components. The generative story is quite similar to what is often used in topic modelling. The most inserting part is to use the document feature to generate the mixture weight. Nevertheless, given the rich literature on mixture modelling, the idea is not surprising, instead, the novelty is rather limited. My further comments/concerns are as follows:
    * Figure 1 shows different relationships between labels, which is quite illustrative to a certain degree. However, It is not clear to me how exactly the mixture model can capture those different types of label dependencies. For me, the proposed model is just a simple mixture model.
    * In eq (2), should $y_{nk}$ be $y_{nl}$?  According to the model, it seems that the labels are generated from a mixture of latent components, I.e., $\pi\times \theta_{1:K}$. So, is $z_{nk}$ associated with each label of a document? Otherwise, it is not clear how each label is generated given the mixture weight $\theta$
* The deviation of the poster inference using variable augmentation is neat. I have a question regards to eq (7), which could be a typo. There are two terms in $exp\{\}$. I wonder if $f^{(k)}_n$ should be outside of the parenthesis in the first term?
* The model makes use of the predicted label covariance to compute the uncertainty for selecting the next sample to be annotated. It is good to be aware of methods like Bald and BatchBald, etc. An in-depth analysis of how the proposed sampling method capture both uncertainty is necessary.
* To compute the acquisition function used in active learning, the authors proposed to use the predicted label covariance in Eq (13), based on the pointe estimated of the mixture distribution $\pi$. However, it is good to further carry out in-depth analysis of how the proposed sampling capture the uncertainty that could linked to the expected error reduction. There is a fairly rich literate of how to quantifying uncertainty in active learning, like Bald, BatchBald [3], Weight MOCU [2], etc.
* The proposed method is evaluated with both synthetic dataset and real-world dataset, which is good.
    * If the major contribution of this paper is also about the proposed mixture model. It is necessary to carry out a set of experiments where the proposed model is compared with the existing multi-label learning models, to say how accurate it is.
    * The reason why labels with frequency less than 20 are removed. However, section 4.1 seems to say that the proposed model can deal with few-shot labels.
    * Should one consider the ranking based measures? Those measures are used more commonly for multilevel learning, see http://manikvarma.org/downloads/XC/XMLRepository.html
    * It seems that the proposed model does not perform well as some other competitors in figure 7 when the size of the acquired sample is relatively small, and on the NUS-wide data set. Any explanation?
    * And presumably, the acquisition size, or called batch size is one at each iteration.

References

[1] Chu et al 2018, Deep Generative Models for Weakly-Supervised Multi-Label Classification
[2] Zhao et al 2021, Uncertainty-aware active learning for optimal Bayesian classifier
[3] Kirsch et al 2019 BatchBALD: Efficient and diverse batch acquisition for deep Bayesian active learning

--- After Review Feedback ---
I acknowledge that the author's response to this reviewer's original comments has been reviewed and considered in the updated rating.

**Time Spent Reviewing:**

>5

---

> ### Author Response · Authors · 2021-08-10
> **Response to Reviewer Fw2q**
>
>
> We would like to thank the reviewer for the valuable and thoughtful
> comments. We summarize our response as follows.
>
> **Q1: The proposed multi-label learning model is basically a simple
> mixture model.**
>
> We leverage the basic concept of a mixture model in the proposed
> GP-B$^2$M model to formulate the label mixture components with the
> primary purpose to ensure the good interpretability of the discovered
> label correlations. However, it is important to note that a simple
> mixture model has a very different design goal and application setting
> than the proposed multi-label active learning model. In general, mixture
> models in their standard form are usually designed to analyze
> large-scale unlabeled data for unsupervised learning (e.g., clustering
> analysis and topic modeling). As a result, they cannot be used to
> predict labels and there is no easy way to directly use them to quantify
> the predictive uncertainty for active sampling, as what our work aims to
> achieve. As stated in the paper, our main contribution is to seamlessly
> integrate features and multi-labels under a unified Bayesian learning
> framework, aiming to discover meaningful label correlations from limited
> labels to better support active sampling. By encoding the inductive bias
> from data features as Bayesian priors, the proposed model jointly learns
> from both labels and features. Such a learning paradigm, which goes far
> beyond a simple mixture model, is both novel and essential to tackle
> highly sparse labels for effective multi-label AL. In addition to the
> unique AL model design, we also develop an innovative end-to-end
> posterior inference algorithm to handle the non-conjugate mapping
> function that connects data features to the mixture components.
> Efficient posterior inference is essential to support real-time
> interactions with human annotators as part of the AL process. Finally, a
> principled sampling function is designed that integrates uncertainty
> from both the data features and labels for effective data sampling. In
> summary, while we leverage the concept of mixture models to ensure good
> interpretability of the learned mixture components, the proposed model
> is designed with a very different purpose and tackles fundamentally
> different research challenges than a simple mixture model. As a result,
> the overall design of the model, the posterior inference algorithm, and
> the sampling function are all different from a simple mixture model.
>
> **Q2: It is not clear to me how exactly the mixture model can capture
> those different types of label dependencies.**
>
> The mixture components essentially capture the co-occurrence
> relationship among a set of related labels. Since this relationship can
> be directly interpreted from the discovered mixture components, we can
> further conduct a fine-grained analysis among multiple components to
> uncover more complex dependencies among the labels. One such example is
> provided in Figure 1 and Lines 64-76, which shows that a complex
> hierarchical dependency among 4 labels $E1$-$E4$ can be successfully uncovered from three discovered mixture components.
>
> **Q3: In eq (2), should $y_{nk}$ be $y_{nl}$**
>
> Thank you for pointing out this typo. The correct log likelihood function
> under our prior setting (Eq. (2) in the main paper) should be:
>
> <$$\ln p({\bf Y}|{\bf X})=\ln \int \int \sum_{Z}\prod_{n} \prod_{k} \prod_{l} p( {\bf f}^{(k)}| {\bf X})p(\theta_{kl})  p(z_{nk}| {\bf f}_n)  p(y_{nl}|z_{nk},\theta_{kl}) dF d\Theta$$>
>
> The above equation also illustrates how the labels are generated given
> the mixture model.
>
> **Q4: In eq. (7), $f_n^{(k)}$ should be outside of the parenthesis in
> the first term?**
>
> Yes, thank you for pointing this out. We will fix it in the revise
> paper.
>
> **Q5: An in-depth analysis of how the proposed sampling method capture
> both uncertainty is necessary; it is good to further carry out in-depth
> analysis of how the proposed sampling capture the uncertainty that could
> linked to the expected error reduction.**
>
> Our sampling strategy shares the same goal as most uncertainty-oriented
> active acquisition approaches. In particular, our model aims to identify
> the unlabeled instances that can cause the most confusion when making a
> prediction. In our model, the predictive distribution integrates the
> uncertainty from both the data features and the labels. The first term
> is the (approximate) predictive entropy, which captures the label
> uncertainty while encoding the label correlations through the label
> covariance matrix. The second term is the predictive variance, which
> captures the uncertainty from the data features. Both terms can be
> efficiently evaluated to support real-time data annotation. Different
> from our model, BALD and batchBALD focus on the mutual information
> between the unknown output and the model parameters. Although BALD can
> be extended to non-parametric models like Gaussian process based
> classification in the binary case, it is not clear how it can be applied
> to the multi-label settings. Furthermore, the second term of its
> acquisition function, which is the expectation of entropy over the
> posterior distribution, is challenging to evaluate. The ELR and
> batchBALD models focus on the parametric Bayesian models. Extension to
> the multi-label setting with correlated labels is non-trivial.
> Furthermore, the computational cost would be potentially high as model
> retraining is required to evaluate on each unlabeled data instance.
> Finally, the DSGM model is based on a sequential generative process and
> focuses on the weakly supervised problem, instead of multi-label active
> learning as in our work.
>
> **Q6: Compare the proposed model with existing multi-label classification models.**
>
> As mentioned in our response to Q1, the proposed work focuses on
> multi-label active learning, instead of mixture model or general
> multi-label classification. The unique design of the model that
> systematically integrates data features and labels aims to accurately
> learn label correlations from very limited labels to support effective
> active sampling. It is also worth to note that general multi-label
> classification models are designed to learn from sufficient labels,
> instead from very limited labels as in the AL setting. The passive
> learning results are presented in Section D.4. We observe that the
> performance of the compared models are typically much lower than ours
> when trained with limited data. As a result, they are less suitable in
> the active learning setting.
>
> **Q7: The reason why labels with frequency less than 20 are removed**
>
> It is worth to note that we only remove extremely rare labels, whose
> occurrences may not provide sufficient statistical support for either
> model training or testing purpose. These removed labels only occur less than 0.5% in their corresponding dataset. This preprocessing trick is
> also commonly used in other multi-label based active learning works
> (e.g., \[1\] and \[2\])
>
> \[1\] Multi-Label Prediction via Compressed Sensing, NIPS 2009.
>
> \[2\] Fast direct search in an optimally compressed continuous target
> space for efficient multi-label active learning, ICML 2019
>
> **Q8: Should the ranking based measures be considered?**
>
> We find the rank-based measurements, such as precision and F score at
> $k$ do not work well for a large and highly imbalanced label space,
> especially for those relatively rare labels. In order to assign labels,
> a threshold usually needs to be determined for each label. Such a
> threshold may vary among labels and between different baseline models,
> which may lead to an unfair comparison. To further verify the AL
> performance of the proposed model, in addition to ROC-AUC, we also
> provide the average precision analysis as an alternative measurement and
> report the result in Figure 10 in Appendix D.3.
>
> **Q9: Does not perform as well when the size is small.**
>
> The relatively low performance in the early phase of AL is mostly due to
> the stronger emphasis on the exploration behavior of the sampling
> function, which usually leads to a longer term gain instead of
> short-term improvement. As AL goes, the proposed model quickly catches
> up and exhibits a much faster improvement, which justifies the benefit
> of early exploration.
>
> **Q10: And presumably, the acquisition size, or called batch size is one
> at each iteration.**
>
> Yes, the proposed active sampling method chooses one sample at each
> step. Thank you for pointing that out and we will clarify this in the
> revised paper.

---

### Official Review · Reviewer_bnVb · 2021-07-19

**Rating:** 6
**Confidence:** 5

**Summary:**

The paper proposes a novel Gaussian Process-Bayesian Bernoulli Mixture (GP-B2M) model to achieve cost-effective sampling for multi-label active learning.


**Limitations And Societal Impact:**

All the labels collected are440assumed to be accurate

**Main Review:**

The proposed method encodes label correlations using a Bayesian Bernoulli mixture of label clusters. Furthermore, a predictive Gaussian Process (GP) is adopted to learn a distribution of mixture coefficients that connect data features with the label clusters.

The learning process is divided into two phases. In the first one, there is a prediction of a probabilistic assignment of the mixture components by learning a distribution of latent functions. In the second one, the predicted mixture assignments are used to refine the parameters of the beta distributions.

The description of the learning method is detailed and well described in Section 3.

As regards the experimental results, Figure 7 could be integrated with a quantitative result. In particular a table could be included reporting the numerical results. Furthermore, a statistical test could be done in order to prove that the results are statistical different.

The adopted single score (AUC) could be integrated using others common scores.



**Time Spent Reviewing:**

3

---

> ### Author Response · Authors · 2021-08-10
> **Response to Reviewer bnVb**
>
>
> We would like to thank the reviewer for the valuable and thoughtful
> comments. We summarize our response as follows.
>
> **Q1: Figure 7 could be integrated with a quantitative result. In
> particular a table could be included reporting the numerical
> results.**
>
> Thank you for the suggestion. We summarize the numerical result in terms
> of mean and standard deviation at different stages of active learning in
> the following tables.
>
>
> |           | NUS-WIDE   |            |            |
> |-----------|------------|------------|------------|
> |           | 100        | 250        | 400        |
> | $GP-B^2M$ | 83.7+/-0.5 | 85.5+/-0.4 | 85.6+/-0.4 |
> | MMC       | 83.1+/-0.5 | 84.1+/-0.5 | 84.3+/-0.7 |
> | ADAPT     | 83+/-0.4   | 84.1+/-0.4 | 84.4+/-0.3 |
> | CS-GP     | 83.4+/-1.3 | 85.7+/-0.8 | 85.6+/-0.8 |
> | CVIRS     | 79.6+/-0.5 | 80.6+/-0.1 | 80.5+/-0.3 |
>
> |           | Enron      |            |            |
> |-----------|------------|------------|------------|
> |           | 100        | 250        | 400        |
> | $GP-B^2M$ | 88.4+/-1   | 89+/-0.5   | 89.2+/-0.3 |
> | MMC       | 85.9+/-0.7 | 88.3+/-0.4 | 89.5+/-0.2 |
> | ADAPT     | 85.9+/-0.6 | 87.3+/-0.5 | 88+/-0.4   |
> | CS-GP     | 86.6+/-1   | 87.6+/-1   | 88+/-0.4   |
> | CVIRS     | 84.8+/-0.3 | 85.9+/-0.2 | 86.1+/-0.1 |
>
> |           | Col5k      |            |            |
> |-----------|------------|------------|------------|
> |           | 100        | 250        | 400        |
> | $GP-B^2M$ | 73.5+/-0.2 | 74.7+/-0.1 | 74.8+/-0.5 |
> | MMC       | 73+/-2     | 72.9+/-0.3 | 73.2+/-0.1 |
> | ADAPT     | 72.7+/-1   | 72.8+/-2   | 73.2+/-1   |
> | CS-GP     | 73.2+/-0.8 | 74.5+/-1   | 74.8+/-0.5 |
> | CVIRS     | 71.3+/-0.7 | 71.9+/-1   | 72.6+/-0.9 |
>
> |           | BibTex     |            |            |
> |-----------|------------|------------|------------|
> |           | 100        | 250        | 400        |
> | $GP-B^2M$ | 60.3+/-0.8 | 62.5+/-0.5 | 63.2+/-1   |
> | MMC       | 58.1+/-1   | 58.4+/-2   | 58.8+/-3   |
> | ADAPT     | 58+/-1.3   | 60.2+/-1.5 | 60.7+/-0.5 |
> | CS-GP     | 58.9+/-0.9 | 61.8+/-0.7 | 62.1+/-2   |
> | CVIRS     | 59+/-0.7   | 59.5+/-0.4 | 58.8+/-0.5 |
>
>
>
> **Q2: a statistical test could be done in order to prove that the
> results are statistical different.**
>
> Following the reviewer's suggestion, we have included the p-values of
> the paired t-test comparing the AUC-ROC of our model and other baselines
> below. The result is reported at different stages of active learning
> $(N=100,250,400)$. The results show our model is significantly better
> than the competitors in most cases expect for early stages of active learning when the
> performance highly depends on the random initialization. We want to
> thank the reviewer for the suggestion but we also notice that
> statistical tests are relatively expensive for evaluating incremental
> learning processes such as active learning.
>
>
> |        | NUS-WIDE  |          |          |
> |--------|-----------|----------|----------|
> |        | N=100     | N=250    | N=400    |
> | MMC    | 0.006     | 0.001    | 0.04     |
> | ADAPT  | 0.005     | 0.001    | 0.01     |
> | CS-GP  | 0.017     | 0.05     | 0.74     |
> | CVIRS  | 1.00E-05  | 1.00E-06 | 1.00E-05 |
>
> |        | Enron     |          |          |
> |--------|-----------|----------|----------|
> |        | N=100     | N=250    | N=400    |
> | MMC    | 0.05      | 0.01     | 0.21     |
> | ADAPT  | 0.006     | 0.007    | 0.01     |
> | CS-GP  | 0.07      | 0.03     | 0.01     |
> | CVIRS  | 0.006     | 1.00E-04 | 1.00E-05 |
>
> |        | Col5k     |          |          |
> |--------|-----------|----------|----------|
> |        | N=100     | N=250    | N=400    |
> | MMC    | 0.47      | 1.00E-04 | 0.003    |
> | ADAPT  | 0.12      | 0.27     | 0.11     |
> | CS-GP  | 0.5       | 0.18     | 0.22     |
> | CVIRS  | 1.00E-04  | 1.00E-04 | 0.008    |
>
> |        | BibTex    |          |          |
> |--------|-----------|----------|----------|
> |        | N=100     | N=250    | N=400    |
> | MMC    | 0.007     | 0.03     | 0.009    |
> | ADAPT  | 0.017     | 1.00E-06 | 0.008    |
> | CS-GP  | 0.02      | 0.23     | 0.11     |
> | CVIRS  | 0.007     | 1.00E-05 | 1.00E-04 |
>
> |        | Delicious |          |          |
> |--------|-----------|----------|----------|
> |        | N=100     | N=250    | N=400    |
> | MMC    | 0.0007    | 0.0001   | 0.02     |
> | ADAPT  | 0.0007    | 0.0001   | 0.001    |
> | CS-GP  | 0.014     | 0.016    | 0.06     |
> | CVIRS  | 0.001     | 0.053    | 0.02     |
>
>
>
> **Q3: AUC could be integrated using others common scores.**
>
> In addition to the AUC-ROC scores as reported in the main paper, we also
> conducted a fine-grained analysis on the label-wise performance, which
> helps to achieve a deeper insight on the performance gain. In
> particular, we compute the average precision improvement rate (referred
> to as API) of individual labels. Figure 10 in Appendix D.3 shows that
> GP-B$^2$M performs significantly better than BRMs on labels with a
> moderate or low frequency, which further confirms the overall good
> performance of GP-B$^2$M as shown in the main paper.

---

### Official Review · Reviewer_vZcB · 2021-07-19

**Rating:** 7
**Confidence:** 4

**Summary:**

The authors propose a multi-label active learning approach leveraging a Gaussian process Bayesian Bernoulli mixture model to model label correlation. Each pattern of label correlation is modeled as a Bernoulli mixture. The Gaussian process predicts the coefficients of the mixture component as a means to obtain an estimate the labels of a given data sample. End-to-end posterior inference is possible due to the introduction of an auxiliary-latent-variable variational inference algorithm. The number of mixture components is estimated duting Bayesian inference. Also, uncertainty estimates for predictions produce label covariance matrix that quantify individual label correlations. Experiments on real-world multi-label datasets demonstrate the effectiveness of the proposed approach.

**Ethical Concerns:**

None.

**Limitations And Societal Impact:**

Line 74: Component 1 -> Component 2

Note that the realization of the GP in (1) has zero mean function whereas in Figure 2(b) is specified as m_k (though not identified as the posterior mean function).

In (7) z_nk - v_nk f_n^k is more likely to be ( z_nk - v_nk ) f_n^k

Why there is a lambda in (6) and (7) when the formulation directly depends on nu, which blocks lambda?

**Main Review:**

The paper is well written, easy to follow and well motivated both from an application and methodology perspective. Though the main data augmentation strategy resulting in a conjugate auxiliary variational Bayes learning is taken from elsewhere ([24] in the paper), its combination with the mixture model is interesting and sufficiently justified. The experiments, including the ablation study, passive learning and active sampling time (the last two in the SM) are extensive and convincing.

**Time Spent Reviewing:**

2.5

---

> ### Author Response · Authors · 2021-08-10
> **Response to Reviewer vZcB**
>
>
> We would like to thank the reviewer for the valuable and thoughtful
> comments. We summarize our response as follows.
>
> **Q1: Note that the realization of the GP in (1) has zero mean
> function whereas in Figure 2(b) is specified as $m_k$ (though not
> identified as the posterior mean function).**
>
> We thank the reviewer for identifying this inconsistency. We will fix
> the notation in the revised paper.
>
> **Q2: Why there is a $\lambda$ in (6) and (7) when the formulation
> directly depends on $\nu$, which blocks $\lambda$?**
>
> Thank you for pointing out this notation issue. The reviewer is correct
> that $\nu$ blocks $\lambda$ so $z_{nk}$ does not depend on $\lambda$.
> This can also be read from the graphical model presented in Figure 3. We
> will update the equation accordingly.

---

### Official Review · Reviewer_9ST8 · 2021-07-22

**Rating:** 7
**Confidence:** 2

**Summary:**

The paper proposes a generative mixture model for multi-label classification which could handle complex label dependencies. First, the paper introduces mixture components with multi-class probabilistic assignment, which contain information about label distribution. Second,  to maximize joint likelihood the method uses variational inference with auxiliary variables. Third, an active learning strategy is proposed based on the predicted label covariance. The method is compared with other active learning algorithms on multi-label classification datasets.

**Limitations And Societal Impact:**

The algorithm cannot be evaluated on the large datasets for multi-label classification because of time complexity, but this is a common problem for all methods of such type.

**Main Review:**

Modeling label distribution is an actively studied problem in multi-label learning research. Because of the internal complexity of the problem, current baseline algorithms for multi-label active learning are quite limited, and can be trained and evaluated only on multi-label datasets of small dimensions (instances/labels/features). This paper makes a good step in improving such algorithms, providing a method for modeling complex label distributions.

The main idea of learning mixture components in a multi-class way using a gaussian process over data points looks quite natural: it helps to break down the method into two steps (learning components, generating label distributions). The introduced auxiliary variables for  variational inference help to come up with an adequate and interpretable algorithm for active sampling. As a disclaimer, I was not able to check all the math behind the variational inference.

The proposed method is compared with state-of-the-art baselines for multi-label active learning by three main criteria: 1) active learning AUC, 2) passive learning AUC, 3) time complexity for active sampling. While being inferior to simpler baselines (CBM) in terms of time complexity, or showing similar performance to GS-GP in the passive learning setting, the method demonstrates good overall performance.

Typos: line 27 -- (BRMs), -> (BRMs); line 74: Component 1 -> Component 2 (?); line 191: digmma -> digamma.

-------
Post-Rebuttal: after reading other reviews and author replies, I decided to keep my score.



**Time Spent Reviewing:**

6

---

> ### Author Response · Authors · 2021-08-10
> **Response to Reviewer 9ST8**
>
>
> We would like to thank the reviewer for the valuable and thoughtful comments. We will also improve the presentation as suggested by the
> reviewer.

---

### Decision · Program_Chairs · 2021-09-28

**Decision:**

Accept (Poster)

**Comment:**

This paper proposes a Gaussian process Bayesian-Bernoulli mixture model and its variational inference algorithm for multiple label active learning and shows its usefulness by experiments including ablation studies. This paper has three positive reviews and one negative one. In response to the author's FB, no consensus was reached in the discussion among the reviewers. An important point of discussion is whether the proposed mixture model is novel and valid. At least for the latter, the experimental comparison shows its superiority over traditional active learning on multiple label classification datasets. As for the former, as the reviewer pointed out, the use of the Bernoulli mixture model as label distribution is natural and not a particularly novel idea. The novelty of the individual techniques themselves is subtle, but the modeling as a whole is reasonable and can be evaluated as a contribution to the advancement in the research field.

**Consistency Experiment:**

NeurIPS has a long history of experimentation. In 2014, NeurIPS ran an experiment in which 10% of submissions were reviewed by two independent committees to quantify the randomness in the review process. This year, we repeated a variant of this experiment to see how the quality of the review process has changed over time.  This paper was part of the experiment and was therefore assigned to two committees (consisting of reviewers, an Area Chair, and a Senior Area Chair) that reached independent decisions.  If both committees made the same recommendation, this recommendation was followed. If a single committee recommended acceptance, the paper was accepted (with the exception of a few cases in which the other committee identified what we considered a fatal flaw, e.g., an error in a key result).

This copy’s committee reached the following decision: **Accept (Poster)**

The other committee assigned to the paper recommended **Reject**.  You can find the other set of reviews, along with any follow up discussion with the authors here:
https://openreview.net/forum?id=ptw0Soe8W8B